# Relationships of Motor Changes with Cognitive and Neuropsychiatric Features in FMR1 Male Carriers Affected with Fragile X-Associated Tremor/Ataxia Syndrome

**DOI:** 10.3390/brainsci12111549

**Published:** 2022-11-15

**Authors:** Darren R. Hocking, Danuta Z. Loesch, Paige Stimpson, Flora Tassone, Anna Atkinson, Elsdon Storey

**Affiliations:** 1Developmental Neuromotor & Cognition Lab, School of Psychology and Public Health, La Trobe University, Melbourne, VIC 3086, Australia; 2School of Psychology and Public Health, La Trobe University, Melbourne, VIC 3086, Australia; 3Psychology Department, Monash Health, Clayton, VIC 3068, Australia; 4Department of Biochemistry and Molecular Medicine, M.I.N.D. Institute, School of Medicine, University of California Davis Medical Center, University of California, Davis, Davis, CA 95616, USA; 5Department of Medicine (Neuroscience), Alfred Hospital Campus, Monash University, Melbourne, VIC 3068, Australia

**Keywords:** FMR1 premutation, CGG repeats, FXTAS, motor scores, cognitive assessments, psychiatric measures, relationships, cerebellar cognitive affective syndrome

## Abstract

The premutation expansion of the Fragile X Messenger Ribonucleoprotein 1 (FMR1) gene on the X chromosome has been linked to a range of clinical and subclinical features. Nearly half of men with FMR1 premutation develop a neurodegenerative disorder; Fragile X-Associated Tremor/Ataxia Syndrome (FXTAS). In this syndrome, cognitive executive decline and psychiatric changes may co-occur with major motor features, and in this study, we explored the interrelationships between these three domains in a sample of adult males affected with FXTAS. A sample of 23 adult males aged between 48 and 80 years (mean = 62.3; SD = 8.8), carrying premutation expansions between 45 and 118 CGG repeats, and affected with FXTAS, were included in this study. We employed a battery of cognitive assessments, two standard motor rating scales, and two self-reported measures of psychiatric symptoms. When controlling for age and/or educational level, where appropriate, there were highly significant correlations between motor rating score for ICARS gait domain, and the scores representing global cognitive decline (ACE-III), processing speed (SDMT), immediate memory (Digit Span), and depression and anxiety scores derived from both SCL90 and DASS instruments. Remarkably, close relationships of UPDRS scores, representing the contribution of Parkinsonism to FXTAS phenotypes, were exclusive to psychiatric scores. Highly significant relationships between CGG repeat size and most scores for three phenotypic domains suggest a close tracking with genetic liability. These findings of relationships between a constellation of phenotypic domains in male PM carriers with FXTAS are reminiscent of other conditions associated with disruption to cerebro-cerebellar circuits.

## 1. Introduction

Fragile X-Associated Tremor/Ataxia Syndrome (FXTAS) is one of the most severe late-onset movement disorders caused by a specific change in the major Fragile X Messenger Ribonucleotide 1 (FMR1) gene [1,2]. It occurs in approximately 45% of older males carrying the FMR1 alleles containing small trinucleotide expansions (55–200 repeats) in the non-coding section, termed ‘premutations’ [3]. The core diagnostic features of FXTAS include kinetic tremor; gait ataxia; and white matter disease in the Middle Cerebellar Peduncles (the MCP sign), seen via Magnetic Resonance Imaging (MRI) [4,5]. White matter disease in the splenium of the corpus callosum has more recently been included amongst the other core FXTAS features [6]. In addition, other changes contributing to this diagnosis (minor criteria) include cognitive decline, seen in the later stages of FXTAS and neuropathy [7,8], and other MRI findings such as global brain atrophy and white matter disease [6,9,10,11], especially in the basis pontis, as well as around the lateral ventricles and in deep white matter of the cerebral hemispheres. Another not infrequent manifestation is Parkinsonism, which may account for some cases of successful surgical treatment typically applied in Parkinson’s disease [12,13].

At the genetic level, there is the co-elevation of FMR1 mRNA as a function of the increased CGG repeat expansion within the premutation range in blood [14], as well as in intranuclear inclusions in neurones and astrocytes [15,16]. This elevation has been linked to a hypothesized pathogenetic mechanism involving a toxic gain-of-function of this transcript, resulting in neuronal death [17,18]. This, and other postulated mechanisms related to CGG expansion and leading to progressive age-dependent neurodegenerative changes in some, but not all, PM carriers, have been reviewed in detail previously [19]. Apart from the obvious effect of age, the other major known factor associated with the occurrence of FXTAS in PM carriers, specifically age of onset of motor signs and motor dysfunction, is the size of the CGG repeat [20,21,22]. However, this syndrome has been linked to a wider range of repeat expansions, including rare instances of the occurrence of FXTAS in male carriers of intermediate (Grey Zone, GZ) range of 41–54 [23,24].

Apart from predominant motor features of kinetic tremor and gait ataxia, cognitive decline (which becomes evident at a later stage of the disorder) is another feature that initially affects the areas of executive functioning, working memory, and information processing speed [25,26,27]. Less extensive studies of psychiatric changes that may occur in FXTAS males reported clinically significant psychiatric symptoms, with elevated anxiety being the most consistent finding across both the Symptom Checklist-90-Revised and the Neuropsychiatric Inventory [28]. Similar types of psychiatric symptoms (assessed in the Symptom Checklist-90-Revised: SCL-90-R) have also been reported in non-FXTAS (or pre-FXTAS) male carriers, albeit at a lower frequency than in those with FXTAS [29]. 

Given that tremor and gait ataxia have been linked to discernible white matter pathology, with the most prominent changes in cerebellar peduncles [30], aberrant connectivity within cerebro-cerebellar circuits might underpin cognitive (mainly executive) dysfunction, as well as emotional and psychiatric changes in male carriers with FXTAS. Indeed, this is consistent with a constellation of features specifically attributed to cerebellar lesions in some other conditions, including the spinocerebellar ataxias in which all three phenotypic domains are affected [31,32]. It is therefore likely that cerebellar damage in FXTAS could give rise to other phenotypic features beyond its well-known role in motor dysfunction.

However, the occurrence of isolated pathological changes, or co-occurrence of changes from more than one domain (motor, cognitive, and psychiatric) provide only limited evidence for a common underlying type and location of pathological changes. Thus, the co-occurrence should be distinguished from the relationships between these domains, where, if statistically interdependent, a disease feature from one domain becomes predictive of the occurrence of another, which indicates that they stem from the same pathogenic mechanism. In this study, we examined the existence of such relationships involving changes in three phenotypic domains—motor, cognitive, and psychiatric—that have been quantified in a sample of adult FMR1 premutation male carriers with diagnosable FXTAS, where the changes in individual domains are clinically observable. 

The results, providing evidence for significant relationships across all three domains in the presence of cerebellar damage, support the hypothesis that the damage to the cortico–ponto–cerebello–thalamo–prefronto–cortical loops can be implicated in the complex clinical and behavioural changes seen in our sample of FMR1 premutation male carriers affected with FXTAS. 

## 2. Materials and Methods

### 2.1. Participants

This study is part of a larger genotype–phenotype relationships project including a total of 40 adult male carriers of the FMR1 premutation alleles aged between 42 and 80 years (mean = 62.3; SD = 8.75). Participants were originally recruited through fragile X families’ referrals from the Victorian Genetic Counselling Clinic of the Murdoch Children’s Research Institute, or from one of several neurology clinics associated with the University of Melbourne and Monash University; the minority (some residing in the other Australian states) were self-referred by postings in the community through the Australian Fragile X Association. Sixteen PM carrier males from this cohort were already included in our earlier publication, where basic cellular metabolism parameters were correlated with white matter lesion burden [33], and the remaining 24 participants had been included in two separate studies: the relationship between AMPK and clinical and genotypic measures [34,35], and also in a comparison of motor and cognitive progression between male and female premutation carriers [36]. For the current study, we used only the data available from the 23 premutation carriers (aged from 48 to 80 years) who met the (revised) diagnostic criteria for FXTAS [37]. Except for one East Asian (Chinese) male, all participants were white Caucasian, mainly of Northern European origin. The remaining 17 non-FXTAS carriers have only been used in Figure 1, where the distribution of the size of CGG repeat expansion for male with FXTAS is shown against this distribution for the non-FXTAS category carriers. The size of expansion of CGG trinucleotide repeat in the FMR1 locus in this combined group ranged from 45 to 118. The reason why our range extends beyond the accepted premutation threshold is that one of our participants affected with FXTAS was a carrier of the intermediate size allele. All participants provided informed consent for the present study, according to protocols approved by the La Trobe University Human Research Ethics Committee (HEC01-85 and HEC15-058).

### 2.2. Neurological Motor and Cognitive Measures

Two standard motor ratings with established inter-rater reliabilities [38,39,40]: the Unified Parkinson’s Disease Rating Scale Part III-Motor (UPDRS-III) [41], and the International Cooperative Ataxia Rating Scale (ICARS) [38] were applied to assess Parkinsonism, and kinetic tremor/gait ataxia, respectively. These were conducted by two neurologists (DZL and ES) with experience in the use of these scales. 

General cognitive functioning was assessed using Addenbrooke’s Cognitive Examination Test Third Edition (ACE-III) [42]. The Similarities and Matrix Reasoning subtests of the Wechsler Adult Intelligence Scale (Third Edition; WAIS-III) [43] provided measures of verbal and non-verbal reasoning, respectively, and were used to calculate Pro-Rated IQ. WAIS-III Digit Span Backward was employed as a measure of working memory [43]. The Symbol Digit Modalities Test (SDMT) was also used as a measure of information processing speed, which underlies other executive functions [44]. 

### 2.3. Psychiatric Symptom Measures

The Symptom Checklist-90-Revised (SCL-90-R) [45] is a 90-item self-report questionnaire providing a measure of a broad range of relevant psychological symptom clusters occurring over the past week. The measure is clustered into nine primary symptom dimensions and a summary score—the Global Severity Index (GSI)—providing a measure of overall psychological distress—is calculated from the average of the primary symptom scales [45]. For the purposes of the current study, we selected a priori Depression, and Anxiety specific symptom domains identified as elevated in previous studies in premutation carriers. We report T-scores for both the symptom dimension scales and the overall level of psychiatric disturbance (GSI T-score), with a score between 60 and 63 considered borderline and >63 classified as above clinically significant threshold.

The Depression Anxiety Stress Scale (DASS-21) is a 21-item self-report shortened version of the original 42 item questionnaire [46]. It consists of a set of self-reported Likert scales to evaluate the severity of psychological symptoms associated with three negative emotional states: depression, anxiety, and tension (stress). Each of the three DASS-21 domains consists of seven questions with the answers rated from zero to three, where each question is categorized as mild, moderate, or severe.

### 2.4. FMR1 Molecular Measures

Genomic DNA was isolated from peripheral blood lymphocytes using standard methods (Purygene Kit; Gentra, Inc., Minneapolis, MN, USA). For Southern blot analysis, 10 micrograms of isolated DNA were digested with EcoRI and NruI. Hybridization was performed using the specific FMR1 genomic dig-labelled StB12.3 probe as previously described [47]. Genomic nDNA was also amplified by PCR [48]. 

### 2.5. Statistical Analysis

Robust regression was used to assess the relationships of each cognitive, motor, and neuropsychiatric score with CGG repeat size, with the adjustment for age and years of education applied wherever significant. The robust regression was also used to perform analysis of the relationship between each individual motor score as predictors, and cognitive and neuropsychiatric scores as outcome variables. Although there was no influence of extreme outliers in our sample, robust regression was used to minimize the effect of any influential observations when present. All the above analyses were conducted using software STATA, version 16.0 (http://www.stata.com). 

## 3. Results

Descriptive statistics of all the measures and scaled scores applied in the analysis of relationships in our sample of FXTAS patients are provided in Table 1. The average age reflects the fact that FXTAS is an old age condition; the average CGG repeat size in this FXTAS sample is 87.7, which is slightly above the mid-point of the premutation range of 55–200 (see this Table 1 and Figure 1). The average value for the ICARS scale score of 27.9 in this sample, which is substantially above average [39], reflects the presence of core features of tremor and gait ataxia in the clinical profile of FXTAS, while the UPDRS score of 17.7 reflects a significant contribution of Parkinsonism in this syndrome [38]. As also shown in Table 1, the reduction in ACE-III scores accompanied by a decline in processing speed reflect obvious cognitive decline in males with FXTAS when compared with normative values.

In contrast, the overall level of psychiatric symptomatology, represented by the GSI T mean score on the SCL-90 and the depression and anxiety scores was only mildly elevated; although the elevation of depression score on the DASS scale (indicative of increased depressive symptomatology) is limited because of the high variability of this measure.

Relationships between each cognitive, motor, and psychiatric symptom score and CGG expansion size (Table 2) show that all the motor scale scores, and the overwhelming majority of cognitive and psychiatric scores, are highly correlated with CGG repeat expansion size.

The results of relationships between motor, cognitive, and psychiatric symptom domains are presented in Table 3. Both the ICARS Total and ICARS kinetic domains show highly significant relationships with nearly all psychiatric scores, while the ICARS Gait domain is highly (negatively) correlated with global cognition (ACE-III), processing speed (SDMT), immediate memory/attention (Digit Span Forwards), and non-verbal reasoning (MRsc) scores. It is of special interest that the UPDRS, which is contributing, but not a major component of FXTAS, showed close associations with all neuropsychiatric test scores included in this analysis.

## 4. Discussion

This study is the first to explore interrelationships between all three phenotypic domains—motor, cognitive, and psychiatric—in a sample of male carriers of FMR1 premutation with a clinical diagnosis of FXTAS. The scores of these three domains were closely intercorrelated, with a range of neuropsychological measures that tap into global cognition, short-term memory, and processing speed. Here, we also provide evidence for the relationship of the motor disease features with the psychiatric symptoms of anxiety and depression, assessed across two independent self-reported measures—the SCL-90-R and the DASS. It is of special significance that we were also able to demonstrate that the features from all three domains were strongly associated with CGG expansion size, which links this constellation to the common primary genetic cause. 

The interrelationships between the cognitive and the psychiatric domains, in the presence of motor involvement in males with FXTAS, is therefore of considerable interest in the context of a description of the constellation of impairments originally termed the Cerebellar Cognitive Affective Syndrome (CCAS). This syndrome comprises cognitive impairment (particularly affecting executive functions, and visuospatial and linguistic abilities), associated with emotional blunting and disinhibition [31]. The topographic cerebellar representation of these impairments are in lobules VI and VII of the posterior lobe, which comprise the cognitive cerebellum, and the posterior vermis, which encompasses the limbic cerebellum, while the anterior lobe and lobule VIII of the postural lobe contain the representation of sensorimotor cerebellum [56]. Regarding the affective component of the constellation of deficits seen after cerebellar damage, it has been proposed that vermis lesions disrupt the connectivity of the cerebro-cerebellar–limbic-cerebellar reciprocally connected loops, and thus affect the regulation of affect and emotions [57].

The interrelationships occurring in FXTAS are of further interest in the context of the cerebellar damage involving the cardinal feature of white matter intensity in the middle cerebellar peduncles. This radiological feature, termed the ‘MCP sign’, is consistently manifested in males affected with this disorder [30]. Indeed, this major (diagnostic) feature was present in all FXTAS carriers included in this study. Although cognitive and psychiatric impairments considered in the CCAS have been linked to the damage to individual, discrete regions of the cerebellum, our findings of the relationships between the cognitive and psychiatric domain features with the scores for cerebellar motor dysfunction, could clearly be attributed to diffuse involvement of the middle cerebellar peduncles. This is because these peduncles carry the fibres from the cerebral cortical areas concerned with sensorimotor processing, as well as the cerebral association areas representing limbic structures, through cerebral–pontine–cerebellar afferent tracts [56]. 

There is accumulating evidence showing that these peduncular MRI changes may occur prior to the age of onset of rapid clinical decline typical of conversion to FXTAS. Consequently, in non-FXTAS male PM carriers, there is evidence for subtle white matter alterations in the MCPs without obvious clinical features [58]. Negative correlations have also been reported between reduced structural connectivity of superior cerebellar peduncles and increased CGG repeat size [59]. We also reported correlations of the size of the CGG expansion with tremor/ataxia motor scores and infratentorial white matter hyperintensities in male PM carriers, with and without FXTAS [34,60]. 

One specific finding of interest in our FXTAS sample relates to the relationships between the posture and gait disturbances subscale from the ICARS, and most of the cognitive deficit scores, especially those assessing global cognition, immediate memory, and processing speed. This is reminiscent of earlier results, which showed associations between reduced cerebellar volume and prolonged step initiation, gait abnormalities, and increased postural sway in male PM carriers, both with and without FXTAS [61,62,63]. Another series of studies showed that subtle FXTAS-specific cognitive impairments co-occurred with postural and gait abnormalities in these two categories [64,65]. Moreover, there is evidence of interdependence between cognition (particularly executive function measures) and gait and balance in healthy older people, as well as in neurodegenerative disorders, including Parkinson’s disease [66,67]. Together, these data suggest that a subtle degree of interdependence between cognitive executive processes and posture/gait abnormalities occurring with aging is magnified by the contribution of cerebellar pathology. 

The current study, which provides novel albeit preliminary data, is not without limitations. First, the small sample size did not permit more complex multivariate regression models to include all three phenotypic domains in the same analysis or applying a correction for multiple testing. Moreover, because of the low power of testing, we may have missed some significant relationships or encountered some false positives. However, the high significance and consistency of our regression results do allow valid conclusions to be drawn from the current study. These have opened a new avenue of research, to explore further potential associations between FMR1 premutation cognitive neuropsychiatric changes and cerebellar motor features utilizing larger samples and more sophisticated imaging techniques. Second, the reliance on self-reporting scales of psychiatric symptoms using SCL-90-R and DASS could be subject to biases or inaccuracy of recall. Finally, the range of cognitive tests has been limited, since this study relied on the available data from a major project involving all premutation carriers. 

In summary, our findings reveal several relationships across motor, cognitive executive, and psychiatric symptom domains in male carriers of FMR1 premutation alleles diagnosed with FXTAS, which, alongside features of cerebellar ataxia and the core peduncular damage, is reminiscent of impairments commonly seen in other disorders involving disruption to cerebro-cerebellar pathways. More specifically, these interrelationships indicate a close interdependence between gait disturbances and cognitive decline in this syndrome. Our findings draw attention to the involvement of psychiatric symptoms in these relationships, especially depression and anxiety, which also suggest the need for psychological intervention and relevant medications to be considered, together with the treatment of motor impairments, in males with FXTAS.

## Figures and Tables

**Figure 1 brainsci-12-01549-f001:**
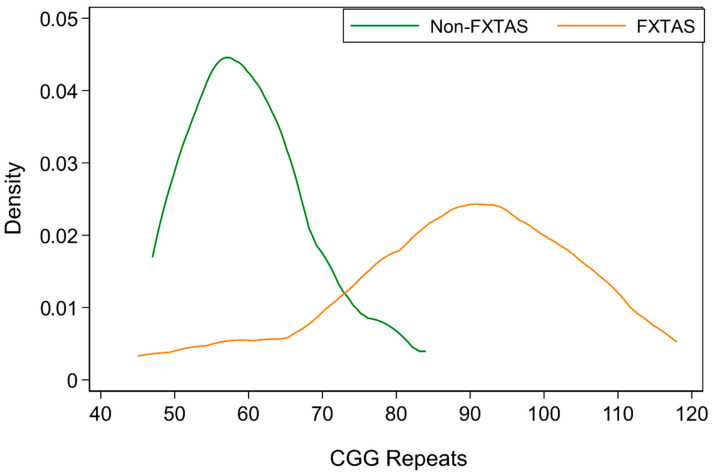
Kernel distribution of the CGG repeat expansion size in 23 male PM carriers affected with FXTAS against this distribution for 17 carriers in the non-FXTAS category.

**Table 1 brainsci-12-01549-t001:** Demographic, motor, and cognitive psychiatric characteristics.

				Normative Values ^1^	
Variable	Mean	SD	Range	Mean	SD
Characteristic					
Age	64.6	7.7	48–80		
Years of Education	12.2	3.8	6–22		
CGG repeats	87.7	17.5	45–118		
Motor Scores					
UPDRS	17.7	13.4	2–39	1.9	2.0 ^2^
ICARS GAIT	9.3	4.7	2–22	2.0	2.0 ^3^
ICARS KINETIC	15.3	7.7	6–29	1.8	1.9 ^3^
ICARS Total	27.9	13.8	10–57	4.1	2.2 ^3^
Cognitive scores					
ACE-III Total	73.6	14.7	48–94	95.7	3.3 ^4^
MR SS	10.0	4.0	3–18	10.4	2.9 ^5^
Pro-rated IQ	97.1	16.4	69–131	-	-
DS Backwards	5.5	1.7	3–10	3.1	1.2 ^6^
SDMT	29.3	14.7	16–59	42.3	8.1 ^7^
Neuropsychiatric scores					
SCL-90-R Depression	59.1	13.9	38–81	49.0	11.2 ^8^
SCL-90-R Anxiety	53.1	13.9	40–62	46.0	11.2 ^8^
SCL-90-R GSI	56.8	11.6	34–81	50.5	8.6 ^8^
DASS Anxiety	7.2	6.7	0–24	6.3	7.0 ^9^
DASS Depression	11.5	11.8	0–36	4.7	4.9 ^9^

^1^ Normative data derived from relevant studies where available. ^2^ Postuma et al. (2012) [49]. ^3^ Fitzpatrick et al. (2012) [50]. ^4^ So et al. (2018) [51]. ^5^ Harrison et al. (2014) [52]. ^6^ Leung et al. (2011) [53]. ^7^ Kiely et al. (2014) [54]. ^8^ Gossett et al. (2016) [55]. ^9^ Henry and Crawford (2005) [46]. ACE-III Total = Addenbrooke’s Cognitive Examination III; MR SS = WAIS-III Matrix Reasoning Scaled Score; DS Backwards = WAIS-III backwards digit span subtotal; SDMT = Symbol Digit Modalities Test.

**Table 2 brainsci-12-01549-t002:** Relationships between each cognitive, motor, and neuropsychiatric score with CGG repeats using robust regression.

	FXTAS				
	*N*		Coef.	Se	*p*-Value
Cognitive scores					
ACE-III Total		16	−0.325	0.130	**0.012**
MR SS		19	−0.085	0.027	**0.001**
* DS Backwards		22	−0.022	0.021	0.301
PRO-rated IQ		18	−0.250	0.110	**0.023**
SDMT Score		17	−0.287	0.082	**<0.001**
Motor scores					
ICARS TOTAL		17	0.308	0.082	**<0.001**
ICARS GAIT		17	0.104	0.027	**<0.001**
ICARS KINETIC		17	0.138	0.060	**0.022**
UPDRS		16	0.288	0.082	**<0.001**
Neuropsychiatric scores					
DASS Anxiety		12	0.115	0.032	**<0.001**
DASS Depression		12	0.240	0.043	**<0.001**
SCL-90-R Depression		11	0.376	0.126	**0.003**
SCL-90-R Anxiety		13	0.212	0.127	0.096
SCL-90-R GSIT		11	0.171	0.025	**<0.001**

* Adjusted for age and year of education whenever significant. ACE-III Total = Addenbrooke’s Cognitive Examination III; MR SS = WAIS-III Matrix Reasoning Scaled Score; DS Backwards = WAIS-III backwards digit span subtotal; SDMT = Symbol Digit Modalities Test; ICARS = International Cooperative Ataxia Rating Scale; SCL-90-R = The Symptom Checklist-90-Revised; GSIT = Global Severity Index; FXTAS = Fragile X-Associated Tremor/Ataxia Syndrome. The bold indicates significance.

**Table 3 brainsci-12-01549-t003:** Relationship between each cognitive or neuropsychiatric (outcome) and motor (predictor) score for the FXTAS group.

	ICARS Total				ICARS Gait			ICARS Kinetic			UPDRS		
	N	Coef.	SE	*p*	Coef.	SE	*p*	Coef.	SE	*p*	Coef.	se	*p*
Cognitive scores													
ACE-III Total	18	−0.42	0.26	0.105	−2.09	0.32	**<0.001**	−0.36	0.50	0.468	−0.39	0.33	0.243
MR.Sc ^1^	18	−0.02	0.07	0.743	−0.20	0.22	0.348	0.01	0.09	0.953	−0.05	0.08	0.525
DS Backwards	18	−0.02	0.03	0.392	−0.14	0.07	**0.050**	−0.01	0.05	0.813	−0.01	0.03	0.859
PRO-rated IQ ^1^	17	−0.02	0.21	0.908	−0.56	0.83	0.499	0.16	0.35	0.643	−0.06	0.39	0.883
SDMT ^2^	16	−0.05	0.24	0.843	−1.52	0.43	**<0.001**	0.29	0.31	0.355	−0.24	0.29	0.399
Neuropsychiatric scores													
DASS Anxiety	12	0.21	0.04	**<0.001**	0.50	0.13	**<0.001**	0.44	0.14	**0.001**	0.28	0.06	**<0.001**
DASS Depression	12	0.62	0.18	**0.001**	1.67	0.22	**<0.001**	1.11	0.46	**0.017**	0.66	0.18	**<0.001**
SCL90 Depression	11	0.61	0.20	**0.002**	1.61	0.49	**0.001**	1.27	0.44	**0.004**	0.68	0.22	**0.002**
SCL90 Anxiety	11	0.49	0.21	**0.017**	0.76	0.53	0.149	1.11	0.33	**0.001**	0.62	0.17	**<0.001**
SCL90 GSIT ^1^	10	0.35	0.18	**0.047**	0.70	0.39	0.071	0.87	0.55	0.116	0.38	0.13	**0.005**

SE: Standard Error. The bold indicates significance; Adjusted for ^1^ years of education; ^2^ age and years of education.

## Data Availability

The data will be made available by the corresponding author upon reasonable request.

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
