# Peer review of "Relationships of Motor Changes with Cognitive and Neuropsychiatric Features in FMR1 Male Carriers Affected with Fragile X-Associated Tremor/Ataxia Syndrome"

_brainsci, 2022, doi:10.3390/brainsci12111549_

Round 1

Reviewer 1 Report

This manuscript addresses the relationship between executive, psychiatric, and motor features in FXpm men with FXTAS. The sample comprised of 23 men with FXTAS, aged 48-80, although there appeared to be significant missing data with some analyses based on a sample as low as 11. Rich direct assessment and self report data evaluated the domains of interest. Findings revealed significant associations between the symptom domains and well as with CGG repeat, but are limited by low sample size and multiple uncorrected comparisons that increase rate of false discovery. Given these sample limitations I have concerns about the reproducibility of this research. I also had some specific questions about hypotheses and interpretation.

Please include a citation for this statement: “Apart from the obvious effect of age, the only other known factor associated with the occurrence of FXTAS in PM carriers is the size of the CGG repeat, with 80-90 repeats (representing the middle of the premutation range) associated with the highest risk for this syndrome.” My understanding is that risk for FXTAS/motor features is related to higher CGG lengths, such as Kraan et al. 2014 or Tassone et al 2007. Also, the wording should be softened or revised for accuracy as age and CGG are not the “only” known risk factors for FXTAS. For example Silva et al. 2013 showed that the APOE e4 allele is associated with 12-fold increased risk for FXTAS; Hong et al. 2022 show that education is a significant neuroprotective factor; I believe that there is also accumulating evidence that smoking increases risk.

This sentence seems to be missing a word: “Apart from predominant motor features of kinetic tremor and gait ataxia, cognitive 65 decline (which becomes evident at a later stage of the disorder) initially affects the areas 66 of executive functioning, working memory and information processing speed…”

In the last sentence of the introduction, this sentence seems out of place and would be better fitting in the discussion: “The results, providing evidence for significant relationships across all three domains, are consistent with damage to the cortico-ponto-cerebello-thalamo-prefronto-cortical loops, which have been implicated in the complex clinical and behavioural changes seen in our sample of FMR1 premutation male carriers affected with FXTAS. Further, we note that this observed constellation of features clustering around this triad of domains is reminiscent of other conditions associated with lesions involving the cerebellum.”

Specific research questions and hypotheses are not stated.

Because the author’s interpretations of the study results infer damage to cerebro-cerebellar circuits, this concept should be better introduced in the introduction. For example, the authors mention other conditions characterized by cerebro-cerebellar circuit disruption that are reminiscent of the premtuation. What are they and how are the symptoms similar? What do we known about brain structure and function in FXTAS/the FXpm and are these circuits implicated?

Also, it is unclear how the findings of this paper, specifically, provide evidence of disrupted crebro-crebellar circuits. Impairments in mother, psychiatric, and executive skills in FXTAS are well documented. Is the inter-relationship between the different symptom domains (psychiatric, motor, executive) that is the focus of this paper what is reminiscent of disrupted cerebro-cerebellar circuits specifically? 

“The occurrence of isolated pathological changes, or co-occurrence of changes from more than one domain (motor, cognitive, and psychiatric) should be distinguished from the relationships between these domains, where, if statistically interdependent, a disease feature from one domain becomes predictive of the occurrence of another.” ---This sentence of the introduction is very interesting and I think gets at the author’s core rationale from the study. However, the sentence is also contains several embedded clauses and is hard to follow. Would it be possible to clarify or elaborate on this idea? Are the authors trying to distinguish from motor, cognitive, and psychiatric changes that co-occur but stem from different mechanisms (and therefore would not be correlated) vs motor, cognitive, and psychiatric changes that stem from the same mechanism (and therefore would be correlated)? I am not sure that I am following this line of reasoning. Because motor, cognitive, and psychiatric changes all stem from FXTAS brain lesions/atrophy --- won’t they always be correlated to a certain extent?  

Figure 1 is confusing. Who are the non-FXTAS participants represented? Are these participants from a different study?

Why was robust regression used? Were the variables normally distributed? Outliers? Were model assumptions upheld?

How is the problem of multiple, uncorrected comparisons accounted for?

In the results the authors state “…and that the average CGG repeat size (see this 160 Table 1 and Figure 1) corresponds to the highest risk for developing this syndrome.” This statement is unclear/misleading. Do the authors mean that that risk for FXTAS was the highest at the “average” CGG repeat size—which was 87 in this sample?  I don’t think this conclusion can be drawn from these data. Certain CGG repeat sizes within the premutation range occur more frequently in the population in general--- for example, CGGs at the lower premutation range are much more common than those in the 140-200 range, which is consistent with the present sample that only has CGG repeats up to 118 represented. Without a similar number of cases representing each CGG repeat length across the premutation range I don’t think that we can conclude from a density plot that the risk for FXTAS was highest at 87 (in other words, if there was only 1 person in the sample with a CGG of ~120 and 5 in the sample who had CGGs at ~80, the density plot is going to be higher at ~80 simply because there are more cases represented in the sample). What I think the only conclusion that can be drawn here is that the average CGG repeat length of the men with FXTAS in this sample was 87.

The total sample size is reported as 23 in the abstract but in Table 2 n’s are as low as 11-13 for the psychiatric scores and most other analyses were based on an n of 17. Where is the data for the remaining participants? It is unclear what conclusions can be drawn from n’s of 11; low sample sizes are associated with low reproducibility and high likelihood that the sample is not representative of the larger population.

With regards to the UPDRS measure, what is the specificity and sensitivity of this measure? Is there evidence that people with non-PD motor disorders would not endorse these items? Without evidence of specificity I am not sure if the finding that only the UPDRS measure related to psychiatric tests is compelling. It may be the case that the UPDRS was related to the psychiatric measures, and not the direct motor measures, because of shared method variance related to both the psychiatric and UPDRS forms being self-report tools.

Were there outliers or cases with high influence (cook’s d) for the CGG associations?

CGG repeat length was truncated in this sample, with the highest CGG repeat represented being 118 when the premutation range is 55-200. This is not surprising given the small sample size but should be acknowledged as it has substantial implications for interpretation (e.g., cannot test or detect curvilinear effects when half of the curve is missing, and curvilinear effects have been observed frequently in studies of women with the FXpm).  

Reviewer 2 Report

Further assessment of motor and cognitive features is a relevant issue in the extended exploration of FXTAS, however in my opinion:

1. Authors could refer to the impact of methods of treatment on the presented features.

Making a Difference-Positive Effect of Unilateral VIM Gamma Knife Thalamotomy in the Therapy of Tremor in Fragile X-Associated Tremor/Ataxia Syndrome (FXTAS). Front Neurol. 2018 Jun 27;9:512. doi: 10.3389/fneur.2018.00512. PMID: 29997574; PMCID: PMC6030249.

Long-term improvement of tremor and ataxia after bilateral DBS of VoP/zona incerta in FXTAS. Neurology. 2015 May 5;84(18):1904-6. doi: 10.1212/WNL.0000000000001553. Epub 2015 Apr 10. PMID: 25862802.

2. The description of the material should be more detailed - comorbidities?

3. Due do the character of the paper and its limitations, this work is more a brief report thank a full research.

Reviewer 3 Report

The paper talks about a sample of 23 adult males aged between 48 and 80 years (mean=62.3; SD=8.8), carrying premutation expansions between 45 and 118 CGG repeats, and affected with fragile X-associated 18 tremor/ataxia syndrome (FXTAS).   The cognitive battery included, have two standard motor rating scales, and two self-reported measures 24 of psychiatric symptoms.   When controlling for age and/or educational level where appropriate, 25 there were highly significant correlations between motor rating score for ICARS Gait domain, and 26 the scores representing global cognitive decline (ACE-III), processing speed (SDMT), immediate memory (Digit Span), and depression and anxiety scores derived from both SCL90 and DASS instruments.   Remarkably, close relationships of UPDRS scores, representing the contribution of parkinsonism to FXTAS phenotypes, were exclusive to psychiatric scores.   Highly significant relationships between CGG repeat size and most scores for three phenotypic domains suggest a close tracking with genetic liability.   These findings of relationships between a constellation of phenotypic domains in male premutation (PM) carriers with FXTAS is reminiscent of other conditions associated with disruption 33 to cerebro-cerebellar circuits.   It is a very well-written paper  with relevant conclusions, more specifically, these interrelationships indicate a close interdependence between gait disturbances and cognitive decline in this syndrome.   Their findings draw attention to the involvement of psychiatric symptoms in these relationships, especially depression and anxiety, suggesting the need for psychological intervention and relevant medications to be considered, together with the treatment of motor impairments, in males with FXTAS.   They also indicate the limitations (but for me there is no problem): First, the small sample size did not permit more complex multivariate regression models to include all three phenotypic domains in the same analysis or applying a correction for multiple testing. Second, because of low power of testing we may have missed some significant relationships or encountered some false positives. The high significance and consistency of our regression results do allow valid conclusions to be drawn from the current study. These have opened a new avenue of research, to explore further potential associations between FMR1- premutation cognitive-neuropsychiatric changes and cerebellar motor features utilizing larger samples and more sophisticated imaging techniques. Second, the reliance on self-reporting scales of psychiatric symptoms using SCL-90-R and DASS could be subject to biases or inaccuracy of recall.  Finally, the range of cognitive tests has been limited, since this study relied on the available data from a major project involving all premutation carriers.

Round 2

Reviewer 2 Report

Authors implemented changes, however the charakter of the work should be changed to brief research report due to the limitations mentioned in the first round.

Author Response

We thank the reviewer for positive comments relating to our response to revisions. However, we do not believe that the study's limitations warrant a brief report, although this decision is ultimately the decision of the assistant editor of the journal.